# Interventional Strategies to Reduce Test Anxiety among Nursing Students: A Systematic Review

**DOI:** 10.3390/ijerph20021233

**Published:** 2023-01-10

**Authors:** Manjit Kaur Khaira, Raja Lexshimi Raja Gopal, Suriati Mohamed Saini, Zaleha Md Isa

**Affiliations:** 1Department of Community Health, Faculty of Medicine, Universiti Kebangsaan Malaysia, Cheras, Kuala Lumpur 56000, Malaysia; 2Faculty of Nursing, University of Cyberjaya, Cyberjaya 63000, Malaysia; 3Department of Psychiatry, Faculty of Medicine, Hospital Canselor Tuanku Muhriz, Universiti Kebangsaan Malaysia, Kuala Lumpur 56000, Malaysia

**Keywords:** test anxiety, nursing students, interventions, systematic review

## Abstract

Nursing students are reported to have moderate to high test anxiety, leading to reduced academic performance, poor self-esteem, and failure to complete the program and practice nursing. This review aims to examine the interventions for test anxiety reduction in nursing students. Following the PRISMA guidelines, peer-reviewed experimental studies published in English between 2016 and 2021 from four databases, EBSCOhost, PubMed, Science Direct, and Scopus, were systematically searched. The findings were presented in tabular and narrative form. Among the 722 studies retrieved, 14 selected studies were critically appraised, guided by the Joanna Briggs checklist for Randomized Controlled Trials and the checklist for Quasi-Experimental Studies, resulting in 11 studies for inclusion in the systematic review. Test anxiety was assessed by different scales. Aromatherapy hand massage, aromatherapy using a diffuser in combination with music therapy, confidence training for test relaxation, coping program, music therapy, emotional freedom technique, animal-assisted intervention, and guided imagery were all found to be effective in reducing test anxiety. In conclusion, while numerous interventions to reduce test anxiety in nursing students were found to be effective, the quality of the studies investigating these interventions was varied with generally small sample sizes and limited follow-up. Future research should be conducted, and the same interventions should be carried out using a larger sample size to strengthen the body of evidence.

## 1. Introduction

University is one of the most important stages in a person’s life, and in today’s educational system, testing and evaluation have become a high priority in determining a student’s future career path. Students react to test and examination pressure with different emotions, but one prominent emotion commonly found in students is test anxiety (TA).

‘Test anxiety’ refers to the set of phenomenological, physiological, and behavioral responses that accompany concern about possible negative consequences or failure on an examination or similar evaluative situation [1]. The severity of TA is related to peer pressure, inability to concentrate, concern about previous examinations, and interpersonal issues [2]. Failure during the examination and the excessive course load [3], unsatisfactory examination preparation, and discomfort with testing methods were some of the major factors influencing test anxiety.

The heightened autonomic response causing symptoms ranging from perspiration and headaches to severe gastrointestinal disturbances and tachycardia are some of the physiological symptoms experiences before an exam [4]. In terms of cognitive symptoms, nursing students who experienced some level of anxiety reported specific cognitive symptoms, such as the inability to recall information and negative self-talk during examination [5]. Anxious students are easily distracted during tests, making it difficult for them to understand instructions.

Nursing education is based on both theory and clinical practice. Tests are an integral part of nursing students’ experience and test anxiety is prevalent among them. Test anxiety (TA) is common in nursing education when students are exposed to assignments and written examinations in the classroom, as well as rating scales, written assignments, e-portfolios, projects on clinical experiences, and examinations, such as the objective structured clinical examination (OSCE) in clinical practice. Test anxiety ranged from moderate to severe among nursing students [6,7,8].

Previous research on TA among nursing students found that it was a significant problem that resulted in negative symptoms that were detrimental to academic success. Poor results, low self-esteem, failure to pass the nursing examination, failure to complete the program and practice nursing [9], and reduced academic performance [10,11] are the outcomes of TA. Exam stress resulted in attentional bias and functional perturbations of a brain circuit that reacted swiftly to test-related threats in highly test-anxious persons [12], resulting in an underestimating of a student’s performance, hence compromising validity and test bias. Mastery over emotions may lead not only to better simulation outcomes, but also to optimal professional performance and improved healthcare quality [13].

Therefore, effective therapeutic interventions that employ complementary alternative methods to reduce test anxiety among nursing students are required to reduce the adverse effects of test anxiety.

While there have been studies conducted on specific interventions to reduce test anxiety, there are only a few systematic evaluations on the topic of test anxiety and interventions. Moreover, only two reviews have examined the effectiveness of complementary alternative techniques for reducing test anxiety among nursing students [14,15,16]. These two comprehensive reviews determined that music therapy, systematic desensitization, hypnotherapy, relaxation training, stress inoculation aromatherapy, cognitive desensitization, and biofeedback-assisted relaxation training had the ability to treat test anxiety [17,18]. However, there have been no further reviews since the year 2017 on the types of interventional therapies to reduce test anxiety among nursing students.

Therefore, this systematic review aims to investigate papers published after the year 2017 to contribute to the body of knowledge by highlighting the different types of alternative therapies that can effectively address TA among nursing students. As the author’s aim is to synthesize primary studies and explore heterogeneity descriptively rather than statistically, a narrative review was carried out. The review is led by the key research question: What interventional strategies have the potential to decrease nursing students’ test anxiety?

## 2. Materials and Methods

Following the identification of the review’s objective, a search strategy was used to identify and retrieve the relevant published studies. The articles were screened and appraised, and data were extracted.

### 2.1. Search Strategy

A preliminary search of the Cochrane Library and the Centre for Reviews and Dissemination was done to access systematic reviews on test anxiety among nursing students before searching a specific database. The search was carried out initially by two reviewers. To find relevant literature, a search strategy was devised. This search strategy was tailored to four databases, EBSCOhost medical collection (MEDLINE, CINAHL, Psychology and Behavioural Sciences Collection), PubMed, Science Direct, and Scopus, using the search terms TITLE-ABS-KEY (“test anxiety” OR “Examination anxiety” OR “Evaluative Anxiety” OR “Evaluation Anxiety”) AND (“Nursing student” OR “student nurse” OR “undergraduate nursing student”) AND (“Interventional strategies” OR “Intervention”). These four major electronic databases were chosen because they are authoritative and include a substantial number of publications in the English language on the subject of test anxiety. They are also commonly used in previously published works. All the searches spanned from databases for publication from the year 2016 to 2021, were journal articles, and are published in English. In addition, the reference to published articles was also searched manually.

### 2.2. Selection Criteria

The selection criteria were based on the Preferred Reporting Items for Systematic Review and Meta-Analysis: the PRISMA 2020 statement: An updated guideline for reporting systematic reviews [19]. The primary goal of the search was to locate the literature on test anxiety in the fields of nursing education, psychology, and behavioral sciences. The search was then narrowed to the field of nursing education. Filters were also used to limit the search to full-text articles published in peer-reviewed journals in English over the last 5 years.

The study eligibility criteria were:Either quantitative or qualitative experimental study;Samples comprised of test-anxious undergraduate nursing students;Psychological or educational interventions or alternative therapies to manage test anxiety;The primary outcome is test anxiety severity measured by scores on any psychometrically reliable and valid self-report instrument;The secondary outcome is the effectiveness of the interventions to manage test anxiety measured by scores on any psychometrically reliable and valid self-report instrument, or through any qualitative data collection and analysis method.

The exclusion criteria for the review include:Studies related to pharmacological trials;Systematic reviews;Articles published in another language;Articles published in other fields (apart from nursing);Articles published before 2016;Articles that include postgraduate nursing students.

### 2.3. Quality Assessment

Studies that met the inclusion criteria were then assessed for methodological quality using the Joanna Briggs Institute (JBI) critical appraisal tool for systematic review. The Joanna Briggs checklist for Randomized Controlled Trials and the checklist for Quasi-Experimental studies were used to appraise the studies [20]. The assessment was undertaken by author 1 and author 2 independently and the studies were selected as “Include” and “Exclude”. The decision to include or exclude was based on the appraisal tool items.

### 2.4. Data Extraction

Data extracted from the included studies were presented in tabular and narrative form using a predefined table. The extracted data included the objective of each study, study design, population (total sample/intervention), instrument, intervention (types and duration), and outcome/results.

## 3. Results

The initial search yielded 722 titles. All articles were gathered into a single folder. Duplicates were removed first. The titles and abstracts were reviewed for relevance by the main author and all authors then independently assessed the eligibility of all potential studies based on their full-text reports.

The number of papers was reduced to 90 after examining the titles and removing duplicates, followed by an additional filtering process focused on publication year and inclusion and exclusion criteria. A review of the abstracts of these papers led to a further narrowing to 40 studies, which the authors then reviewed in depth.

Studies that did not meet the inclusion criteria were then excluded, leaving a total of 14 studies. Discrepancies were discussed among all the authors and after a critical appraisal of the 14 studies, another 3 studies were excluded, leaving a total of 11 studies to be included in the review. The PRISMA diagram (Preferred Reporting Items for Systematic Reviews and Meta-Analyses) [19] depicts the steps in the selection and screening process, as well as the reasons for exclusion. Figure 1 describes the Prisma 2020 Flow Diagram to show search results. Table 1 presents the results of the critical appraisal for RCT studies and Table 2 presents the results of the critical appraisal for quasi-experimental studies.

### 3.1. General Characteristics of the Included Studies

The review included eleven studies from four different countries. There were four from the United States, two from Iran, three from Turkey, and one from Korea. Five of the studies utilized a quasi-experimental design, five used an RCT design, and one was a mixed-method study that used both an RCT design and a qualitative design. All eleven included studies investigated the effectiveness of the intervention on test anxiety, but one of the studies also evaluated the effect of the intervention on self-efficacy [29].

Nine studies investigated the effect of the intervention on written tests, with three of them focusing on the effect of the intervention on the objective structured clinical examination (OSCE). The largest sample size was *n* = 132, while the smallest sample size was *n* = 14. The number of times an intervention was carried out ranged from one to five times, in a period of 10 min to 60 min, spread out over one day to two weeks. In all 11 studies, post-intervention analysis was performed immediately after the intervention, or after the examination. However, none of the 11 studies included post-intervention follow-ups.

### 3.2. Characteristics of the Interventions

The eleven studies were all designed to determine the effectiveness of interventions in reducing test anxiety among nursing students. Three studies compared aromatherapy hand massage to unscented hand massage [29], emotional freedom technique with music therapy [24], and music therapy with aromatherapy and a combination of both [25].

Both [24,29] compared the interventions to control groups, whereas [25] did not compare the interventions to a control group. One study examined the effectiveness of confidence training for relaxation [34], while another looked at a coping program [27].

Two studies investigated the effectiveness of aromatherapy on test anxiety [26,28]. The remaining four studies investigated interventions such as emotional freedom technique [30], music therapy [21], animal-assisted intervention with dog therapy [22], and guided imagery intervention [23]. Six studies compared the level of test anxiety before and after an intervention [21,24,25,28,29,30].

Other studies assessed test anxiety levels at the beginning of the term and then after an OSCE examination [27], before the first examination and then one week after the third examination [34], one day before the examination and one day after the examination [26], and before a mid-semester examination and then just before a final examination [23]. Reference [22] assesses test anxiety at four different points; pre-pre-test at the time of consent (in the middle of the semester), pre-prior to intervention (start of the next semester), post-immediately after the intervention, and post-post immediately after the examination.

Nine of the included study interventions were solely implemented in nursing institutions, whereas the confidence training interventions were also carried out at home by participants on their own time [34]. In only one study, the intervention was completely carried out by participants at their own homes daily for one week before the final examination [23].

### 3.3. Test Anxiety Measuring Instruments

The State-Trait Anxiety Inventory (STAI) [35] was the most commonly used tool to measure test anxiety and was used in five studies [21,22,25,27,30]. The STAI, developed by Spielberger in 1983, consists of 40 items with 20 items pertaining to state anxiety and 20 items measuring trait anxiety. The questionnaire has been translated into many languages. State anxiety items include: “I am tense; I am worried” and “I feel calm; I feel secure”. Trait anxiety items include: “I worry too much over something that really doesn’t matter” and “I am content; I am a steady person”. The intensity of the participants’ feelings is rated on a 4-point Likert scale: 1 (not at all), 2 (somewhat), 3 (moderately so), and 5 (very much so). The total score for each 20 items STAI Form ranged from a minimum of 20 to a maximum of 80. Higher scores denote a higher level of anxiety. The reliability and validity of the questionnaire in the research, including the Korean, Turkish, and Persian versions of STAI, varied between 0.86, 0.95, 0.65, and 0.85, respectively.

Two studies [29,34] used the Westside Test Anxiety Scale (WTAS) [36]. The scale is constructed to measure anxiety impairments, with most items asking directly about performance impairment or about worrying, which interferes with concentration. This instrument involves 10 items on a 5-point Likert scale ranging from 1 (not at all true or never true) to 5 (extremely or always true). Six of 10 items are based on test anxiety, and 4 items are based on worry and dread when taking a test. Each score is tallied, with a possibility of 50 total points, which are then divided by 10 with the results given: 1.0 to 1.9 low test anxiety, 2.0 to 2.5 normal test anxiety, 2.5 to 2.9 high normal test anxiety, 3.0 to 3.4 moderate high-test anxiety, 3.5 to 3.9 high test anxiety, and 4.0 to 5.0 extremely high-test anxiety. As this tool has been used to measure test anxiety in many different studies and has an alpha of 0.78, a split-half reliability of 0.77, and a validity coefficient of 0.51, it was determined to be a well-suited instrument for the studies.

Two other studies [23,28] used the Test Anxiety Inventory (TA Inventory) [37], which consisted of 25 multiple choice questions rated on a four point scale from 0 to 3 (as “Never”, “Seldom”, “Sometimes”, and “Often”, respectively). Thus, the lowest and the highest possible total scores of the inventory were, respectively, 0 and 75, with higher scores showing higher levels of TA.

The Situational-Continuous Anxiety Inventory [24], Beck Anxiety Inventory (BAI) [38], Cognitive Test Anxiety Scale (CTAS) [39], and Revised Test Anxiety Scale (RTAS) [40] were used in the remaining studies. All these instruments contained items to which participants reported on a Likert scale of 1–4 whereby a higher score indicates a higher level of test anxiety. The Cronbach’s alpha value of all these instruments were high, ranging between 0.88 to 0.96. Self-efficacy was measured using the General Self-Efficacy Scale (GSES) [41] in one of the studies [29].

The Subjective Unit of Distress scale (SUD) [42] was used to assess the severity of the participant’s symptoms and as a repeated measure to assess the progress of an intervention [30]. The severity of distress is evaluated by subjects on an 11-point Likert scale. 0 corresponds to absolutely no distress, while 10 corresponds to the maximum possible distress. Cronbach’s alpha and the correlation coefficient for the SUD scale were found to be 0.95 and 0.90, respectively, in this study.

### 3.4. Test Anxiety Interventions

Three of the studies compared aromatherapy hand massage to unscented hand massage, emotional freedom technique with music therapy, and music therapy with aromatherapy and a combination of the two. One study evaluated the effectiveness of confidence training for relaxation, while another looked at a coping program. Two studies investigated the effectiveness of aromatherapy on test anxiety. The remaining four studies focused on interventions like emotional freedom technique, music therapy, animal-assisted intervention with dog therapy, and guided imagery intervention.

Eight studies investigating the effectiveness of the intervention reported statistically significant results. While there was a significant reduction in test anxiety levels after the implementation of aromatherapy hand massage compared to unscented hand massage [29], there was no significant reduction reported in the implementation of aromatherapy of lemon essential oil using a handheld nasal inhaler [26] and through a diffuser [28]. However, aromatherapy using essential oils in a diffuser in combination with music therapy was found to be more effective than either intervention alone [25].

Confidence training for test relaxation, which utilizes active physical exercises that seek to control anxiety and adaptive image which improve attitudes (STARS) [34], and a coping program that included relaxation and soothing techniques, diaphragmatic breathing training, and progressive muscle relaxation training accompanied by light instrumental music [27], were both reported to be significantly effective in reducing test anxiety among nursing students.

Music therapy was found to be effective in one study [24], whereas, in another study there was no significant difference in the outcome [21]. The emotional freedom technique (EFT) was found to be effective in reducing test anxiety in both studies in the review [24,30]. Animal-assisted intervention using dog therapy [22] and guided imagery [23] were all found to be effective in reducing test anxiety.

The qualitative findings were categorized into three main contexts, themes, and subthemes [21]. Anxiety coping methods (positive mindset, listening to music, and spending time with positive-minded friends), students’ perspectives on music therapy (not affecting anxiety about the OSCE, inability to distinguish positive/negative effect, and perceived as a waste of time), and students’ suggestions for reducing anxiety about the OSCE (detailed information about OSCE and more practical applications) were the three main contexts and themes reported. Focus group interviews with students revealed that music was suitable for lowering anxiety regularly, but not before an exam. The details of the test anxiety interventions are summarized in Table 3.

## 4. Discussion

This systematic review’s purpose was to examine the effectiveness of interventions in reducing test anxiety among nursing students. The eleven papers that investigated the effectiveness of therapies on test anxiety in nursing students were found and analyzed. Eight out of the eleven studies reported a statistically significant reduction in the level of TA experienced by nursing students. The findings of the review corroborate those of previous systematic reviews and meta-analyses of TA and its interventions evaluated in various student populations [15,16,17,18,43,44,45].

In contrast to the systematic review conducted by [18], our review did not attempt to examine the contributing factors to test anxiety or the effect that test anxiety has on academic achievement, however, it expands Shapiro’s review in several ways. While Shapiro’s review was from studies dated 2002 to 2012 and included both intervention and non-intervention studies, as well as grey literature and studies that include both graduate and prelicensure students, our review focused exclusively on peer-reviewed studies on interventions for TA in nursing students carried out after the year 2016.

This review suggests that test anxiety is a significant health problem affecting nursing students, as evidenced by studies investigating the effectiveness of an intervention to reduce it. While test anxiety is a universal problem and research on the subject has been conducted extensively throughout Asia, North America, and Europe, only limited studies in a few geographical locations such as Turkey, the USA, Korea, and Iran met the inclusion criteria of this review. This could be due to a limitation in the review inclusion criteria, or it could indicate the need to broaden this research into a global context.

The interventions found to be effective in reducing test anxiety varied, and a few were evaluated in more than one study. Two out of the four studies examining aromatherapy in this review found aromatherapy to be effective in reducing test anxiety. However, aromatherapy was delivered differently in these studies. While aromatherapy delivered through hand massage for 10 min [29] and air diffusion using a lamp for 20 min were both found to be effective in reducing test anxiety, a combination of aromatherapy and music therapy was found to be more significant in reducing test anxiety [25]. Aromatherapy using lavender, on the other hand, administered via humidifier and handheld nasal inhaler did not result in a reduction in anxiety [26,28]. This finding contradicts the findings of a review by [18], who found that aromatherapy, particularly lavender, was effective in reducing test anxiety among nursing students. However, the small sample size in comparison to the larger sample size in Shapiro’s review may have influenced the outcome. These findings also contradicted the findings of the review by [46], where aromatherapy using essential lemon oil diffused in a classroom reduced test anxiety of the nursing students.

Confidence training for the test which utilizes active physical exercises [34] and a coping program that included relaxation and soothing techniques, diaphragmatic breathing training, and progressive muscle relaxation training accompanied by light instrumental music [27], were found to be effective, however, due to the small sample size in this study, larger studies are required to replicate these findings. While a recent systematic review with a larger sample size found exercise to be effective in alleviating students’ test anxiety [16], the studies included in the review were not specifically on nursing students.

The emotional freedom technique (EFT) was found to be effective in reducing test anxiety in both studies in this review [24,30]. The findings of the studies in this review correlated with findings by [47,48] who found EFT to be an effective tool in reducing anxiety and stress among nursing students. A similar finding was found in a review of interventions to reduce stress, anxiety, and depression [49].

While music therapy was found to be effective in reducing test anxiety before an OSCE [24], a focus group interview reported otherwise as they claimed that while music therapy helps to reduce anxiety in their daily lives, it was not effective before an OSCE [21]. However, a reduction in test anxiety was noted when music therapy was used in combination with aromatherapy [25]. As both the studies on music therapy in this review focused only on test anxiety before an OSCE, further studies on the effectiveness of music therapy to reduce test anxiety before examination besides the OSCE are needed as the findings of the review by [18] did indicate that music therapy reduced test anxiety among nursing students. This finding correlated with similar findings among undergraduate students, whereby the findings suggest a benefit of music therapy among nursing students and other undergraduate students experiencing test anxiety [50,51].

Guided imagery and animal-assisted intervention using dog therapy in this review were both found to be equally effective in reducing test anxiety [22,52]. The finding in this review correlated with studies on the effectiveness of guided imagery in reducing test anxiety among nursing students [53] and other categories of students in colleges [54]. Similarly, animal-assisted interventions were found to reduce the anxiety of pharmacy and physical graduate students before a practical exam [31].

An evaluation of the designs, samples, interventions, procedures, tools, analyses, and findings provided in the publications demonstrated good reporting quality in this review. Most studies provided adequate information to allow replication of the interventions, and the effectiveness of the interventions was assessed using a valid and reliable instrument. The State-Trait Anxiety Inventory (STAI), a well-known and established instrument for assessing test anxiety, was used in most of the studies because of its high levels of both internal reliability and validity. However, when completing self-report measures of anxiety, nursing students at each institution may favor the intervention out of a desire to assist instructors or researchers.

The studies were conducted at a nursing institution, which increased their logical validity. Although sufficient statistical analysis was provided in most studies, the absence of power analyses, the smaller sample size in some of the studies, and the restriction of all investigations to single institutions limit the generalizability of the study findings.

The positive effects of intervention strategies on test anxiety may not have lasted if the possible impact of test anxiety symptoms on individuals were directly affected because the therapies in the review were not generally evaluated over an extended period.

## 5. Limitations

This review has several limitations. Firstly, the search was limited to English-language, peer-reviewed research. Excluding published studies that were not peer-reviewed and reported in a different language may have restricted the possibility to provide a more comprehensive assessment of the effectiveness of various types of interventions in reducing test anxiety. Secondly, although there is a large amount of literature on test anxiety, there is limited information about strategies to minimize test anxiety in nursing students, and therefore only 11 studies were selected for the review. The third limitation of this review is that different outcome measurement instruments were utilized in the included studies. A quantitative analysis (meta-analysis of the data) was not possible due to the heterogeneity between studies that resulted from differences in the types of interventions, sample sizes, and scales used.

## 6. Future Direction

Additional rigorous research and long-term monitoring of outcomes are required to corroborate these findings. In the future, larger-sample, higher-quality RCTs will be required to allow the analysis of the results of a meta-analysis. Further research is required that explicitly describes the lived experiences of university students and nursing students in relation to therapies for test anxiety. With this information, a more complete picture of the experience of test anxiety and the success of treatments may emerge. In addition, future research should study the minimal intervention duration and the question of whether a shorter intervention period might ease students’ test anxiety in a timely and effective manner. Furthermore, because test anxiety is a situation-specific condition, future research should also include when post- and follow-up assessments are conducted so that the efficacy of interventions can be better determined. All university students are anticipated to continue experiencing test anxiety in the future. Educators should explore adding interventions to prevent, manage, and minimize test anxiety in students, despite the limited evidence base for each specific intervention.

## 7. Conclusions

It cannot be argued that test anxiety will persist as long as exams are given, however, the issue of test anxiety cannot be disregarded. In light of the changing educational environment and the pressures experienced by nursing students, research to determine effective interventions to cope with test anxiety is essential. This review updated the findings of the review [17,18] in which both aromatherapy and music therapy interventions were found to be effective. This review compared both these interventions with other more recent types of interventions. Although aromatherapy, Confidence Training for Tests Relaxation (STARS), the Emotional Freedom Technique (EFT), a coping program that included relaxation and soothing techniques, diaphragmatic breathing training, and progressive muscle relaxation training accompanied by light instrumental music, and music therapy, were reported to be effective in reducing test anxiety among nursing students, the smaller sample size in some of the studies and the heterogeneity of the results make it difficult to conclude that these interventions are effective.

Our current review updating the findings of [17,18] indicated a lack of empirical evidence of effective interventions to lower nursing students’ test anxiety. This review’s findings suggest the need for additional studies on effective methods to assist nursing students on how to manage test anxiety. While the current use of interventions requires further research to determine the success and gaps, educators need this review of the literature to understand the existence of various types of interventions and how these interventions can help students overcome test anxiety. Furthermore, due to a lack of evidence for long-term effectiveness, we recognize the need for further research on these interventions’ post-implementation.

While this paper could only capture the existing literature, we believe the next stage of literature needs to address broader implications than short-term test anxiety-reducing interventions in order to achieve institutional strategies, such as student well-being and success.

## Figures and Tables

**Figure 1 ijerph-20-01233-f001:**
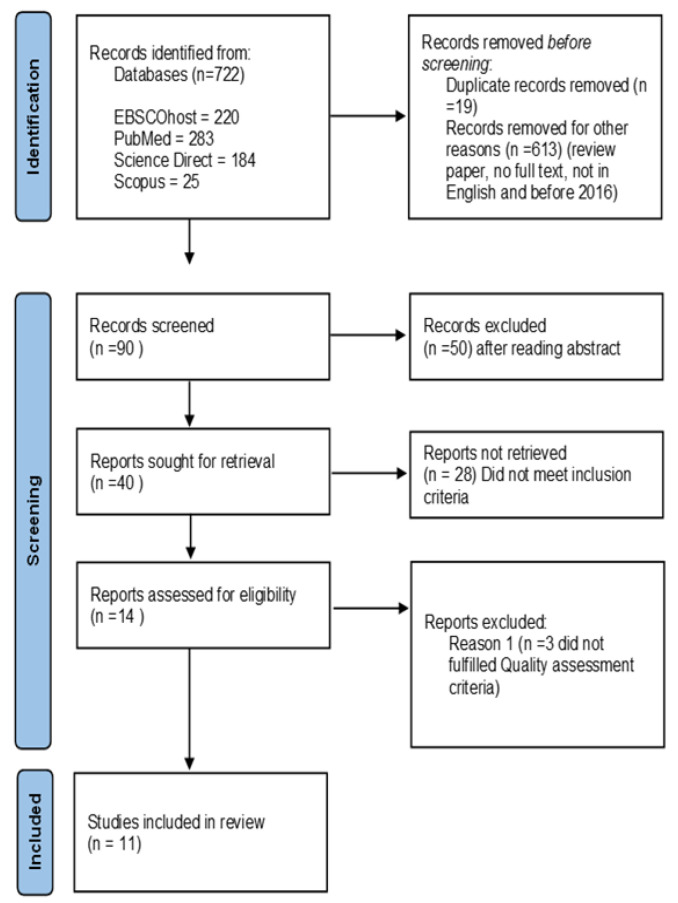
PRISMA flow diagram.

**Table 1 ijerph-20-01233-t001:** Results of the Critical Appraisals for Inclusion or Exclusion of Randomised Controlled Trials.

Study/Q	Q1	Q2	Q3	Q4	Q5	Q6	Q7	Q8	Q9	Q10	Q11	Q12	Q13	Decision
**Eyüboğlu et al., 2021** [21]	Y	U	Y	U	Y	Y	Y	Y	Y	Y	Y	Y	Y	INCLUDE
**Anderson & Brown 2021** [22]	Y	Y	Y	U	U	U	Y	Y	Y	Y	Y	Y	Y	INCLUDE
**Maghaminejad et al., 2021** [23]	Y	U	Y	U	U	U	Y	Y	Y	Y	Y	Y	Y	INCLUDE
**Inangil et al., 2020** [24]	Y	U	Y	Y	U	U	Y	Y	Y	Y	Y	Y	Y	INCLUDE
**Son et al., 2019** [25]	Y	Y	Y	Y	Y	Y	Y	Y	Y	Y	Y	Y	Y	INCLUDE
**Johnson 2019** [26]	Y	U	Y	U	U	U	U	Y	Y	Y	Y	Y	Y	INCLUDE

NOTE., Y = yes, U = unclear.

**Table 2 ijerph-20-01233-t002:** Results of the Critical Appraisals for Inclusion or Exclusion of Quasi-Experimental Design.

Study/Q	Q1	Q2	Q3	Q4	Q5	Q6	Q7	Q8	Q9	Decision
**Mojarrab et al., 2020** [27]	Y	Y	N	Y	N	Y	Y	Y	Y	INCLUDE
**Jafarbegloo et al., 2020** [28]	Y	Y	N	Y	N	Y	Y	Y	Y	INCLUDE
**Farner et al., 2019** [29]	Y	Y	Y	Y	NA	Y	Y	Y	Y	INCLUDE
**Vural et al., 2019** [30]	Y	N	N	N	Y	Y	Y	Y	Y	INCLUDE
**Ward & Smith, 2019** [31]	Y	NA	N	N	U	Y	U	Y	Y	EXCLUDE
**Kolgari et al., 2018** [32]	Y	Y	Y	Y	N	N	Y	Y	Y	EXCLUDE
**Steward et al., 2018** [33]	Y	Y	U	Y	N	Y	Y	Y	Y	EXCLUDE
**Miller et al., 2016** [34]	Y	Y	N	N	N	Y	Y	Y	Y	INCLUDE

NOTE., Y = yes U = unclear, N = no, NA = not applicable.

**Table 3 ijerph-20-01233-t003:** Summary of results of extraction-test anxiety interventions.

Author Info	Aim	Study Design	Participants/Total Sample/ (Control (CG)/Intervention (IG))	Instrument	Intervention/Length	Results Control (CG) Mean ± SD (Pre/Post/*p* Value)/Intervention (IG) Mean ± SD (Pre/Post/*p* Value)
1. Eyüboğlu et al., 2021 [21] Turkey	Determine the effect of music therapy on nursing students’ first objective structured clinical exam (OSCE) success, anxiety levels, and vital signs, and reveal their views about music therapy in the context of an exam	RCT/Qualitative study Pre-test/post-test	A first-year nursing student enrolled in the Fundamentals of Nursing II course in the Nursing Department N = 132 IG *n* = 61 CG *n* = 64 Qualitative phase *n* = 22 (divided into three focus groups)	Informative Features Form (IFF). State-Trait Anxiety Inventory (STAI: Spielberger, 1980)-Turkish version Vital Signs Assessment Form (VSAF). Skill Checklists (SC). Semi-Structured Focus Group Interview Form (1 week after OSCE)	Music therapy 5 sessions (2/per week for 2 weeks before OSCE and one session immediately before OSCE A session lasted for 60 min	No statistically significant difference between the pre-and post-OSCE average anxiety scores of the CG and IG (*p* > 0.05) A significant increase in the anxiety score in both CG and IG after the intervention CG (37.30 ± 3.38/41.39 ± 6.34/*p* = 0.0001) IG (37.57 ± 4.62/41.70 ± 5.45/*p* = 0.0001) A significantly higher pulse in both IG and CG after the OSCE (*p* < 0.05). CG (99.08 ± 17.39/104.52 ± 16.28/*p* = 0.001) IG (97.46 ± 16.22/105.93 ± 16.28/*p* = 0.0001) A significantly lower pre- and post-systolic and diastolic blood pressure values of the CG and IG (*p* < 0.05). Focus group interview—suitable for reducing anxiety in their daily lives, but not before the exam.
2. Anderson & Brown, 2021 [22] USA	Evaluated an animal-assisted intervention using therapy dogs to determine if student anxiety due to a medication calculation exam decreases with exposure to the therapy dogs.	RCT (pre-pre-post-post)	Students enrolled in the Bachelor of Science in Nursing program at a private university. N = 90 students IG *n =* 46 CG *n* = 44	State-trait anxiety inventory (STAI; Spielberger, 1983)	Animal-assisted intervention using therapy dogs for 35–45 min (Before the exam)	A statistically significant difference between intervention and control groups Wilk’s ∧ = 0.761, F (8, 79) = 3.103, *p* < 0.01 Between-subjects tests determined a statistically significant difference for both state F (1,86) = 14.031, *p* < 0.001, partial η2 = 0.140, and trait F(1,86) = 6.647, *p* = 0.012 anxiety between the intervention and control groups at the time of the post-test. There was no statistically significant difference between the intervention and control group TA during the post-post-test (*p* = 0.160)
3. İnangil et al., 2020 [24] Turkey	Determine the effects of music therapy and EFT on situational anxiety and vital signs in nursing students before an OSCE and compare these two methods.	RCT Pre-test/post-test	Nursing student from the Faculty of Health Sciences, Department of Nursing N = 90 IG *n* = 30/30 CG *n* = 30	Student Identification Form Situational Anxiety Scale (Spielberger, 1980) Vital signs (BP, Pulse, and SpO2)	Music therapy (MTG)-an instrumental piece of traditional Turkish music played on the Saz, for 15 min using an MP3 player. Emotional Freedom Technique (EFT) Performed for 15 min following instruction by a certified EFT instructor.	A statistically significant decrease in the median in MTG and EFT groups after intervention (*p* < 0.05). CG (55.00/56.00/*p* = 0.464) MTG (55.00/44.50/*p* = 0.000) EFT (56.50/43.00/*p* = 0.001) There was no statistically significant difference in the anxiety levels between music therapy and EFT groups (*p* = 0.459). A statistically significant decrease in pulse rate in the EFT group and an increase in the SpO2 rate in the music therapy group (*p* < 0.05)
4. Maghaminejad et al., 2020 [23] Iran	Investigate the effects of guided imagery (GI) on TA among the 1st year nursing students	RCT Pre-test/post-test	1st year nursing students in the Faculty of Nursing and Midwifery of Kashan University of Medical Sciences N = 92 IG *n* = 20 CG *n* = 18/20	Test Anxiety Inventory (TA Inventory; Abolghasemi et al., 1996)	30-min Guided Imagery audio file (1 week before the final exam and 1 h daily in the evening	No significant difference in the group in terms of demographic characteristics (*p* > 0.05). CG (44.94 ± 7.34/42.83 ± 13.56/*p* = 0.55) IG (50.50 ± 13.90/33.90 ± 14.39/*p* = 0.003) The post-test mean score of TA in the intervention group was significantly less than the control group (*p* = 0.05)
5. Mojarrab et al., 2020 [27] Iran	Evaluate the effect of an anxiety coping program on the OSCE performance level of first-year nursing students.	Quasi-experimental (pre-test/post-test with the control group)	1st year nursing students at Nursing and Midwifery faculty, University of Medical Science, Shiraz, Iran N = 76 IG *n* = 41 CG *n* = 35	State-trait anxiety inventory (STAI; Spielberger, 1983) (Persian version)	A coping program that included relaxation and soothing techniques, diaphragmatic breathing training, and progressive muscle relaxation training accompanied by light instrumental music. One session before OSCE (40 min)	A significant increase in the anxiety score after the OSCE in the CG whereas there was a significant decrease in the anxiety score after the OSCE in the IG. CG (42.03 ± 9.618/50.349 ± 9.459/*p* = 0.001) IG (47.78 ± 10.492/36.17 ± 12.595/*p* = 0.001) The OSCE results showed an increase compared to the midterm results in the IG compared to the CG group where there was a decline compared to the midterm results CG (8.0694 ± 0.68179/7.7614 ± 0.89470/*p* = 0.001) IG (7.2093 ± 1.05615/8.1580 ± 1.07190/*p* = 0.001) A significant difference in the mean anxiety score between the groups in the pre-and post-exam (*p* < 0.05)
6. Jafarbegloo et al., 2020 [28] Iran	Investigate the effect of inhalation aromatherapy with lavender essential oil on nursing students’ test anxiety	Quasi-experimental (pre-test/posttest with the control group)	Nursing students enrolled in 3rd semester of their 4-year nursing education at Qom University of Medical Sciences. N = 33 IG *n* = 16 CG *n* = 17	Test Anxiety Inventory (TA Inventory; Abolghasemi et al., 1996)	Aromatherapy (10 drops of lavender essential oil added to water and distributed through a humidifier) For 15 min on the day of the exam	A significant decrease in test anxiety scores within the experimental group after the intervention. CG (37.14 ± 15.9/31.7 ± 16.6/*p* = 0.15) IG (38.8 ± 14.73/32.3 ± 15.38/*p* = 0.03) No statistically significant difference was determined between the two groups (*p* > 0.05)
7. Farner et al., 2019 [29] USA	Evaluate the effects of two interventions (Aromatherapy Hand Massage (AHM) and unscented Hand Massage) on test anxiety and self-efficacy.	Quasi-experimental (pre-test/post-test design study with 2 experimental groups and one control group)	Senior-level nursing students enrolled at an urban University undergoing critical care nursing courses. N = 14 AHM group *n* = 4 Unscented HM group *n* = 4 CG (*n* = 6)	Westside TAS (WTAS; Driscoll 2007) General Self-Efficacy Scale (GSES; Sherer et al., 1982)	Aromatherapy Hand Massage (AHM) 4 drops essential oil mixed with 5 mL of carrier oil (10 min) Unscented Hand Massage (HM) (10 min)	Significant improvements between baseline and post-intervention WTAS scores across all groups (*p* = 0.010). No differences in the GSES scores overall or between the groups AHM group saw a 12.4% decrease in test anxiety, compared with only a 1.76% decrease in the control group. AHM group noted a higher percentage change from the mean (5.93%) than both the HM and C groups (−3.03% and 0.52%, respectively) in self-efficacy.
8. Johnson 2019 [26] USA	Assess the effect of inhaled lemon essential oil on Cognitive Test Anxiety (CTA) among nursing students.	RCT Pre-test/post-test	Nursing students enrolled at a public university N = 31 IG *n* = 16 CG *n* = 15	Cognitive Test Anxiety Scale (CTAS; Cassady, 2000)	Aromatherapy (several drops (8–9) of lemon essential oil) using a personal, hand-held nasal inhaler utilized during the exam	No statistically significant difference in the average CTAS score between the pre-test and post-test in the experimental group, t (14) = 2.01; *p* = 0.064 No statistically significant difference in the average change in CTAS scores between the 2 groups: t(29) = 1.35; *p* = 0.19.
9. Son et al., 2019 [25] Korea	Compare the effects of aromatherapy combined with music therapy on test anxiety.	RCT Pre-test/post-test	sophomore female nursing students at a Nursing college N = 107 IG (AG) *n* = 32, IG (MTG) *n* = 32 IG (AMTG) *n* = 34	Revised Test Anxiety Scale (Benson and El-Zahhar 1994, Korean version, Cho 2011) Spielberger State Anxiety Inventory-Y (STAI; Spielberger, 1980), Korean version; Hahn, Lee, and Chon 1996) Numeric rating score (NRS) Performance checklist.	Aromatherapy (two types of essential oils diffused into the air using a lamp for 20 min) Music Therapy (in space equipped with sound system (20 min) Before the test	No significant differences in these characteristics among the three groups (*p* > 0.05). A significant difference among all three groups after intervention for test anxiety (F = 4.29, *p* = 0.016), state anxiety (F = 4.77, *p* = 0.011), stress (F = 4.62, *p* = 0.012), and fundamental nursing skill performance (F = 8.04, *p* = 0.001). Post hoc test-AMTG was associated with a significant decrease in test anxiety, state anxiety, stress, and increased subjects’ fundamental nursing skill performance, as compared with the separate intervention groups.
10. Vural et al., 2019 [30] Turkey	Determine the effects of EFT on exam anxiety in Turkish nursing students	Quasi-experimental (No control group)	2nd year nursing student studying at a university in Istanbul.	State-trait anxiety inventory (STAI; Spielberger, 1980) Beck anxiety inventory (BAI; Beck et al.,1988) Subjective units of distress scale (SUD; Wolpe, 1973)	Emotional Freedom Technique (EFT) before exam 3 × 2 min (first EFT session along with the practitioner)	A significant decrease in SUD score after each EFT session SUD 1(4.81 ± 2.19/4.05 ± 2.27/*p* = 0.005) SUD 2 (4.05 ± 2.18/3.13 ± 2.10/*p* = 0.002) SUD 3 (3.20 ± 2.13/1.98 ± 1.97/*p* = 0.000) No significant difference in BAI mean score post-EFT IG (34.71 ± 11.65/32.95 ± 7.59/*p* = 0.885 A significant decrease in the STAI score after EFT STAI Tx1 (43.56 ± 10.59/42.85 ± 8.55/*p* = 0.003) STAI Tx2 (45.83 ± 6.92/41.12 ± 8.41/*p* 0.000)
11. Miller et al., 2016 [34] USA	Examine the effect of relaxation techniques, Confidence Training for Tests in the reduction of test anxiety in first-semester nursing students	Quasi-experimental (pre-test/post-test. with no control group)	ADN and PN students enrolled in the fundamentals of nursing course in the first semester. N = 16	Westside TAS (WTAS; Driscoll 2007)	Confidence Training for Tests relaxation (STARS) utilizing an audio CD 3 sessions × 30 min each	A significant decrease in the mean test anxiety scores post-intervention IG (3.5 ± 1.053/2.87 ± 1.079/*p* < 0.05) A significant difference in the sum of item responses pre and post-intervention IG (34.5 ± 10.53/28.7 ± 10.79/*p* < 0.05)

## Data Availability

The data presented in this study are available in the list of the references.

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
