# Peer review of "Interventional Strategies to Reduce Test Anxiety among Nursing Students: A Systematic Review"

_ijerph, 2023, doi:10.3390/ijerph20021233_

Round 1

Reviewer 1 Report

Dear authors,

It was my pleasure to review the manuscript entitled “Interventional Strategies To Reduce Test Anxiety Among Nursing Students: A Systematic Review”. However, I have some concerns that need to be clarified, especially related to the method:

-       What type of systematic review was used? More than 14 different types are described in the literature: scoping review, narrative review, umbrella review, etc. A clarification in the text would be of great interest.

-       Why the choice of these databases? A justification in the text would be helpful.

-       What content analysis process was used?

-       It would be interesting to put the limitations section as the last sub-section of the methodology section. In addition, it should be reworded. Of the three paragraphs, the first (lines 352-360) and the last (lines 367-372) are again summaries of the results, which has already been done in the discussion section. Therefore, I recommend writing the limitations section with a focus on limitations only.

-       I consider that the results section 3.2. Test anxiety measuring instruments (lines 202-213) should be expanded, as it is the essence of this article. What is the structure of each of these tests, what are their objectives, is there a reason given in the texts analyzed for their choice, and is there a reason given in the texts analyzed for their choice?

-       I would add a final, independent section after discussion, focused on future lines of research. In the end, a systematic review is not only to show an overview of the research so far, but also a tool that allows, based on the biases found, to show new avenues and paths of research (whether new reviews or empirical processes).

-       Finally, congratulations once again to the authors for a very interesting article. I hope my comments will be helpful.

Sincerely.

Reviewer 2 Report

Dear authors,

Congratulations on your work! You have spotted a gap in the literature and set out to bridge it.

I think you might want to use the plural in all the subtitles (3. Results; 3.4. Test anxiety interventions). Also, in Line 372 use period at the end of the sentence (after "of these findings").

Reviewer 3 Report

The present study made a systematic review about the interventional strategies to reduce test anxiety among nursing students.  The authors came to the conclusions that there are numerous effective interventions to reduce test anxiety in nursing students. However, the authors also pointed out that the quality of the studies investigating these interventions was varied with generally small sample sizes and limited follow-up.

This study used the standard process of systemic review and no obvious flaws was found as to its methodology and presentation. Unfortunately, I think that the literature retrieved in this study is too heterogeneous and the number of included studies is too small. Just as the authors mentioned, there are many intervention methods used in the literature, which make the effect of the each intervention method is not clear. Based on that, the authors' conclusions are not so convincing. I think this is the biggest problem about the present study. 

In addition, given that anxiety is widespread before the exam, it does affect students' academic performance, but why do the authors need to explore the ways to reduce anxiety in nursing students alone? Is there any special significance?
